# The Impact of Intermittent Hypoxia on Metabolism and Cognition

**DOI:** 10.3390/ijms232112957

**Published:** 2022-10-26

**Authors:** Ryogo Shobatake, Hiroyo Ota, Nobuyuki Takahashi, Satoshi Ueno, Kazuma Sugie, Shin Takasawa

**Affiliations:** 1Department of Neurology, Nara Medical University, 840 Shijo-cho, Kashihara 634-8522, Japan; 2Department of Neurology, Nara City Hospital, 1-50-1 Higashikidera-cho, Nara 630-8305, Japan; 3Department of Biochemistry, Nara Medical University, 840 Shijo-cho, Kashihara 634-8521, Japan; 4Department Respiratory Medicine, Nara Medical University, 840 Shijo-cho, Kashihara 634-8522, Japan

**Keywords:** sleep apnea, intermittent hypoxia, appetite, obesity, diabetes, insulin resistance, hypertension, dementia, Alzheimer’s disease, cognitive impairment

## Abstract

Intermittent hypoxia (IH), one of the primary pathologies of sleep apnea syndrome (SAS), exposes cells throughout the body to repeated cycles of hypoxia/normoxia that result in oxidative stress and systemic inflammation. Since SAS is epidemiologically strongly correlated with type 2 diabetes/insulin resistance, obesity, hypertension, and dyslipidemia included in metabolic syndrome, the effects of IH on gene expression in the corresponding cells of each organ have been studied intensively to clarify the molecular mechanism of the association between SAS and metabolic syndrome. Dementia has recently been recognized as a serious health problem due to its increasing incidence, and a large body of evidence has shown its strong correlation with SAS and metabolic disorders. In this narrative review, we first outline the effects of IH on the expression of genes related to metabolism in neuronal cells, pancreatic β cells, hepatocytes, adipocytes, myocytes, and renal cells (mainly based on the results of our experiments). Next, we discuss the literature regarding the mechanisms by which metabolic disorders and IH develop dementia to understand how IH directly and indirectly leads to the development of dementia.

## 1. Introduction

Sleep apnea is a common sleep disorder in which pauses in breathing or periods of shallow breathing occur more frequently than normal during sleep. Each pause lasts from a few seconds to several minutes, and occurs many times during the night. Sleep apnea may be either obstructive sleep apnea (OSA), in which breathing is interrupted by a blockage of air flow, central sleep apnea (CSA), in which periodic involuntary breathing simply stops, or a combination of the two [1]. Sleep apnea syndrome (SAS), which includes OSA, CSA, and a combination of the two, is one of the common forms of sleep disorder and is characterized by recurrent oxygen deprivation during sleep, daytime sleepiness, and decreased quality of life [2]. SAS is often caused by a partial or total closure of the upper airway, leading to a reduction of airflow during sleep, and the prevalence of SAS, defined as an apnea-hypopnea index of 5 or higher, averaged 22% in men and 17% in women in 11 epidemiological studies published between 1993 and 2013 [3]. It is also estimated that nearly one billion of the world’s adults aged 30–69 may contract SAS [4]. During sleep, whether the clinical type is OSA, CSA, or the combination of the two, patients with SAS experience repeated apneas and hypopneas, exposing organs and tissues to the alternation of hypoxia and normoxia, or intermittent hypoxia (IH) [5]. IH generates oxidative stress abnormalities that are similar to those observed in ischemia-reperfusion injury [6,7,8,9,10], and leads to redox-activated signal transduction pathways in inflammation [11,12,13]. SAS is associated with various complications, including obesity [14], type 2 diabetes mellitus (DM) [5,15,16,17], hypertension [18], dyslipidemia [19], which are components of metabolic syndrome, and dementia [20]. While evidence is accumulating on the epidemiological association of SAS with metabolic syndrome and dementia, the molecular mechanisms underlying the onset of metabolic syndrome and dementia due to SAS have not been fully elucidated.

In this narrative review, we summarize the current knowledge regarding the link between IH, one of the major pathological conditions of SAS, and appetite regulation, insulin resistance, and hypertension, including our results of cellular studies and discuss the impact of IH on metabolic syndrome and dementia.

## 2. The Effect of IH on Appetite Regulation in the Gut–Brain Axis

Obesity is recognized as a major public health concern, and its increasing prevalence was about 13% among the world’s adult population (11% of men and 15% of women) in 2016, nearly tripling between 1975 and 2016 [21]. Traditionally, evidence has accumulated concerning the strong association between SAS and obesity [22]. Obesity can cause SAS because of airway obstruction due to airway narrowing induced by an excess of fat tissue around the neck [23]. Quintas et al. indicated a prevalence of 70.5% and 22% of OSA in obese and normal-weight patients, respectively [24]. In contrast, Li et al. revealed that body mass index (BMI) was significantly lower in Far-East Asian men with SAS than in white men with SAS when adjusted for sex, age, and disease severity, and that the mean BMI of Far-East Asian men was below the national norms for men [25]. Eckert et al. also reported that 70% of patients with OSA have one or more non-anatomical phenotypes, such as upper airway muscle lower responsiveness during sleep, low respiratory arousal threshold, or higher loop gain in addition to anatomical factors [26].

Although the etiology of obesity is due to a variety of complex factors, and energy balance is intricately regulated by many neurobiological and physiological mechanisms, increased appetite contributes to weight gain. Excessive appetite is one of the risk factors for obesity. In appetite regulation, hypothalamic neuroendocrine cells control homeostasis via the production and secretion of neurohormones into general circulation [27]. The hypothalamus is an important relay site for integrating signals from central and peripheral pathways in the neuronal circuits to control energy homeostasis [28,29], and is composed of distinct hypothalamic nuclei, including the arcuate nucleus (ARC), the paraventricular nucleus, the lateral hypothalamic area, the dorsomedial nucleus, and the ventromedial nucleus. In the ARC, there are two principal neuronal populations for feeding behavior: the neurons that express orexigenic neuropeptide Y/agouti-related peptide (NPY/AGRP), and those that express anorexigenic proopiomelanocortin/cocaine- and amphetamine-regulated transcript (POMC/CART). Both of them form the central melanocortin system, with downstream target neurons expressing the melanocortin 3 receptor (MC3R) and melanocortin 4 receptor (MC4R) [28]. Intracerebroventricular injection of insulin in rodents has been demonstrated to reduce food intake by downregulating the expression of NPY and AGRP, and by upregulating the expression of POMC and CART in the ARC [30]. NPY/AGRP neurons are inhibited by leptin, insulin, and the enteric hormone peptide YY (PYY)_3–36_, and they are stimulated by ghrelin (GHRL), an orexigenic hormone released from the gastric mucosa [31]. Galanin (GAL) is an orexigenic neuropeptide expressed by a majority of noradrenergic neurons in many tissues throughout the body [32]. Pyroglutamylated RFamide peptide (QRFP) is another orexigenic neuropeptide, produced in human hypothalamus [33]. Galanin-like peptide (GALP) is a neuropeptide responsible for energy homeostasis, discovered in the porcine hypothalamus [34].

Although the detailed mechanism by which IH affects the appetite regulation in patients with SAS has not been fully clarified, we previously investigated the effect of IH on the expression(s) of major appetite-regulating neuropeptide and receptor genes, such as *POMC*, *CART*, *GAL*, *GALP*, *GHRL*, *QRFP*, *AGRP*, *NPY*, and *MC4R*, using human neuronal cells (NB-1, SH-SY5Y, and SK-N-SH) and an in vitro IH system. This is a controlled gas delivery system that regulates the flow of nitrogen and oxygen to generate IH. We also demonstrated that IH significantly upregulates the mRNA levels of *POMC* and *CART*, which are anorexigenic neuropeptides, in human neuronal cells by real-time reverse transcription polymerase chain reaction (RT-PCR). Subsequent promotor assays revealed that the IH-induced upregulation of *POMC* and *CART* mRNA levels is caused by the transcriptional activation of the *POMC* and *CART* genes, and that the −705 to −686 promoter region of the *POMC* gene and the −950 to −929 region of the *CART* gene are essential for IH-induced promoter activity. Moreover, a computer-aided search revealed that both the −705 to −686 promoter region of the *POMC* gene and the −950 to −929 region of the *CART* gene contain possible GATA transcription factor binding sequences. RT-PCR indicated that among GATA family members, *GATA2* and *GATA3* mRNAs were mainly expressed in human neuronal cells. The introduction of human *GATA2* and *GATA3* small interfering RNA (siRNA)s into human neuronal cells abolished the upregulation of *POMC* and *CART* mRNAs induced by IH, showing that GATA2 and GATA3 are essential transcription factors for the IH-induced upregulation of *POMC* and *CART* mRNAs [35]. These results suggest that IH may have an anorexigenic effect on patients with SAS through the transcriptional activation of *POMC* and *CART* via GATA transcription factors in the central nervous system (CNS) [35]. Concerning the significance of GATA factors in the IH condition, Park et al. demonstrated that Gata4 is involved in the IH-induced upregulation of B cell lymphoma 2 (Bcl-2) and B cell lymphoma-extra large (Bcl-xL) in mouse myocardial cells, although the mechanism was unclear [36].

The enteric nervous system (ENS) works together with the CNS as the gut–brain axis, along which neurotransmitters convey information from the gut to the brain through afferent fibers, and from the brain to the gut through efferent fibers to regulate gut secretion and motility [37]. The gut–brain axis transmits various neural and hormonal signals from the gut to the brain via the vagus nerve [38,39,40]. Gut peptides in the gastrointestinal epithelium transmit nutrient-derived energy signals afferently by activating vagal and spinal afferents in ENS neurons to control appetite appropriately [41]. The regulation of feeding behavior requires various circulating gut hormones, such as GHRL, glucagon-like peptide (GLP-1), PYY, and neurotensin (NTS), as well as hypothalamic factors (POMC, CART, GAL, GALP, orexin, NPY, QRFP, AGRP). We previously examined the possibility that IH could have an anorexigenic effect on the ENS, in addition to the CNS. Using human and rodent enteroendocrine cell lines (Caco-2 and STC-1) and the same in vitro IH system, we examined the effect of IH on the gene expression of *PYY*, *GLP-1*, and *NTS*, which are major anorexigenic gut hormones, and investigated the mechanism of their gene regulation in Caco-2 cells treated with IH. This study indicated the IH-induced upregulation of *PYY*, *GLP-1*, and *NTS* mRNA levels in enteroendocrine cells, implying that the appetite of patients with SAS could be decreased even via the ENS [42]. With regard to the gene regulatory mechanism, the promoter activities of *PYY*, *Glucagon* (which encodes a preprotein, part of which is cleaved into GLP-1), and *NTS* were not upregulated by IH. Furthermore, real-time RT-PCR indicated that the levels of microRNA (miR)-96, miR-527, and miR-2116, which degrade *PYY*, *GLP-1*, and *NTS* mRNAs, respectively, in IH-treated cells were not decreased by IH, showing no involvement of miR-mediated posttranscriptional regulation. Therefore, taking into account the possibility that the promoter assays did not reflect the authentic chromatin structure of nuclear DNAs (which can alter the transcriptional efficiency), we treated human enteroendocrine cells with 5-azacytidine (5AZC), genistein, trichostatin A (TSA), resveratrol, and quercetin (which affect the epigenetic regulation of gene expression by modifying the chromatin structure of nuclear DNAs), and revealed that TSA significantly upregulated *PYY*, *GLP-1*, and *NTS* mRNA levels even in the normoxia condition, and that 5AZC significantly decreased *PYY*, *GLP-1*, and *NTS* mRNA levels in the IH condition. In addition, the combined treatment of TSA and 5AZC recovered the IH-induced upregulation of *PYY*, *GLP-1*, and *NTS* mRNA levels [42]. These results imply that the IH-induced upregulation of *PYY*, *GLP-1*, and *NTS* mRNA levels could result from an alteration in the chromatin structure of the genes, that TSA has an effect similar to IH, and that 5AZC has an effect opposite to IH on the expression of *PYY*, *GLP-1*, and *NTS* mRNAs. In this study, we showed the possibility that IH has an anorexigenic effect on patients with SAS by upregulating the expression of *PYY*, *GLP-1*, and *NTS* genes in enteroendocrine cells, and that IH can change the chromatin structure of the *PYY*, *GLP-1*, and *NTS* genes [42].

From the results of our studies, IH upregulates the expression of *POMC* and *CART* mRNAs in neuronal cells and the expression of *PYY*, *GLP-1*, and *NTS* mRNAs in enteroendocrine cells, implying that IH itself may lead to loss of appetite in patients with SAS via the gut–brain axis (Figure 1). Nevertheless, since appetite is only one of the factors that dictate body weight, multifaceted studies targeting the impact of IH on insulin resistance and systemic metabolic abnormalities will continue to be needed to elucidate the mechanism by which IH affects body weight.

## 3. The Effect of IH on Insulin Resistance

SAS is an independent risk factor for the development and progression of type 2 DM [43], and for insulin resistance [44]. The retrospective study also indicated that desaturation parameters assessed by polysomnography examination are associated with an increased risk of type 2 DM [45]. The level of serum hypoxia-inducible factor (HIF)-1α was found to be significantly increased in patients with type 2 DM compared to a control group [46], and cell culture studies showed that HIF-1α regulates both glucose uptake and glycolytic enzyme activity, significantly promoting the process of glycolysis [47]. IH during sleep leads to alterations in pancreatic β cell functions, such as glucose-induced insulin biosynthesis, which includes preproinsulin gene transcription, proinsulin synthesis (translation), and insulin secretion. The progression to type 2 DM depends on impaired glucose-induced insulin secretion from pancreatic β cells and the presence of insulin resistance in peripheral target tissues and organs, including the liver, adipose tissue, and skeletal muscle [7]. IH is reported to cause β cell replication and apoptosis even without hyperglycemia [48], implying a possible mechanism by which IH acts as a β cell replication factor. In fact, Ota et al. have demonstrated that IH significantly downregulates the gene expression of cluster of differentiation (CD)38 (ADP-ribosyl cyclase/cyclic ADP-ribose [cADPR] hydrolase: EC 3.2.2.6) [5], which is essential for glucose-induced insulin secretion through the mobilization of Ca^2+^ from the intracellular Ca^2+^ pool via a type 2 ryanodine receptor Ca^2+^ channel, by cADPR in primary cultured rat, mouse, and human pancreatic islets and animal model experiments [49,50,51,52,53,54,55,56,57]. IH also increased rodent pancreatic β cell replication by upregulating the regenerating gene (*Reg*) family, which encodes autocrine and paracrine growth factors necessary for β cell replication [58,59,60,61], and by upregulating an antiapoptotic hepatocyte growth factor, the upregulation of which may combat the occurrence of β cell dysfunction and insulin resistance [62]. With respect to insulin resistance due to IH in human hepatocytes, Uchiyama et al. revealed that IH stress upregulates the levels of *SELENOP*, which encodes selenoprotein P, a hepatokine responsible for insulin resistance, and upregulates the levels of *hepatocarcinoma-intestine-pancreas/pancreatitis-associated protein* (*HIP*/*PAP*), which encodes HIP/PAP, a type III Reg family member, to proliferate the hepatocytes, through the downregulation of miR-203, leading to the proliferation of liver cells with high levels of *SELENOP* mRNA [17]. Adipose tissue dysfunction has also been suggested to play an important role in dysmetabolism in OSA [63]. In human adipocytes, Uchiyama et al. clearly indicated that the expression of resistin (RETN), tumor necrosis factor-α (TNF-α), and C–C motif chemokine ligand 2 (CCL2), which are bioactive mediators produced and released from adipocytes and are categorized in adipokines, was increased by IH via the downregulation of miR-452 [64]. This implies that the upregulation of RETN, TNF-α, and CCL2 in SAS patients may induce a pro-inflammatory phenotype of adipose tissue, leading to the development of insulin resistance and decreased insulin sensitivity, and that miR-452 could play essential roles in the regulation of these gene expressions. Skeletal muscles also play an important role in insulin-sensitive glucose uptake via glucose transporter 4 (solute carrier family 2, facilitated glucose transporter member 4), although few studies have examined the effect of IH on glucose uptake and metabolism in skeletal muscles. Recently, we indicated that IH exposure increases interleukin (IL)-8, osteonectin (ON), and myonectin (MN), which are all involved in inflammation and glucose metabolism. It also increases mRNA levels in mammalian muscle cells (where octamer binding transcription factor 1 (OCT1) is required for the IH-induced upregulation of IL-8 and MN mRNA expression levels). Here, nuclear factor erythroid 2-related factor 2 (NRF2) serves as a key factor for the IH-induced upregulation of ON mRNA expression [65,66,67]. Additionally, Tang et al. recently reported the possible involvement of IH in selective alterations of the gut microbiota, which may affect the pathophysiological development of type 2 DM [68].

In conclusion, there is accumulating evidence indicating that IH induces impaired glucose tolerance and insulin resistance in pancreatic β cells, hepatocytes, adipocytes, and skeletal muscle cells by upregulating the genes responsible for insulin resistance [69] (Figure 2).

## 4. The Effect of IH on Blood Pressure Regulation

A growing body of evidence indicates that SAS causes hypertension through recurrent episodes of oxygen desaturation and reoxygenation [70]. The prevalence of hypertension in SAS patients is two- to three-fold higher than that in the general population. Furthermore, habitual snoring, one of the hallmark symptoms of SAS, has been reported to be associated with hypertension in several epidemiologic studies [71,72]. It has also been reported that SAS may be a risk factor for gestational hypertension [73,74] and preeclampsia [75,76,77] in pregnant women. The pathophysiology of hypertension in cases of SAS is complex and dependent on various factors. The pathophysiology of hypertension in OSA includes sympathetic tone, endothelial dysfunction, altered baroreceptor reflexes, and the increased renin-angiotensin-aldosterone system (RAS) [70]. RAS plays an important role in the regulation of both extracellular fluid volume and blood pressure. Increasing activity of the RAS contributes to hypertension in SAS patients [78]. Several reports indicate that the components of the RAS, such as renin (Ren), are increased in IH [79]. Ren is synthesized and secreted from the renal juxtaglomerular (JG) cells that are located in the afferent arteriole of the glomerulus, and is regarded as a primary determinant of RAS activity because it accelerates the RAS by converting angiotensinogen into angiotensin I. In addition, several studies have suggested that the regulation of Ren (synthesis and/or secretion) is controlled by the CD38-cADPR-mediated signaling pathway [80]. cADPR serves as a second messenger to mobilize Ca^2+^ from the endoplasmic reticulum via ryanodine receptors [49,51,54,81]. CD38 is a type II glycoprotein that synthesizes cADPR in physiological conditions [50,55,82]. Several studies have described how the CD38-cADPR-mediated signaling pathway is related to the pathogenesis and disease conditions of various diseases [53,55,82]. It has also been reported that the CD38-cADPR-mediated signaling pathway affects the expression of Ren in a prototype of JG cells, namely As4.1 cells [83]. Takeda et al. previously investigated the mRNA expression of both *Ren* and *CD38*, as well as their regulation mechanisms, in Ren-producing cells treated with IH. Human HEK293 and mouse As4.1 renal cells were exposed to IH or normoxia for 24 h, and mRNA levels were measured by real-time RT-PCR. The mRNA levels of *Ren* and *Cd38* were significantly increased by IH in human and mouse renal cells, indicating that they could be involved in the CD38-cADPR signaling pathway. The secreted Ren in the culture medium and the cellular Cd38 were measured by enzyme-linked immunosorbent assay and western blot, respectively, and the medium Ren and cellular Cd38 in the As4.1 JG cells were significantly increased in response to IH. To clarify the mechanism of Ren expression in the As4.1 cells, the *Cd38* gene was knocked down using siRNA. While the expression of *Ren* and *Cd38* were significantly increased in response to IH in the presence of control RNA, the siRNA for Cd38 (*siCd38*) introduction inhibited not only the IH-induced increases in the *Cd38* mRNA, but also the *Ren* mRNA in the As4.1 cells, indicating that the increases in the *Ren* mRNA by IH were caused by the *Cd38* expression. Furthermore, the addition of 8-bromo-cADPR, the cell membrane-permeable cADPR antagonist [54,84], into the As4.1 cell culture medium suppressed the IH-induced increases in the *Ren* and *Cd38* mRNAs, indicating that the increases observed in the *Ren* levels in response to IH were induced by the CD38-cADPR-mediated signaling pathway. Promoter activity of both the *Ren* and *Cd38* genes was not increased by IH; therefore, it was considered that the IH-induced upregulation of both *Ren* and *Cd38* was regulated post-transcriptionally. A computer-aided search using the MicroRNA.org program for the miR targeting both *Ren* and *Cd38* mRNAs revealed that both mRNAs have a potential target sequence for miR-203, and the level of miR-203 in the IH-treated cells was found to be significantly decreased when compared with the normoxia-treated cells. The IH-induced upregulation of *Ren* and *Cd38* mRNAs was abolished by the miR-203 mimic introduction, but was not abolished by the miR-203 mimic non-specific control RNA. These results indicate that IH downregulates miR-203 in Ren-producing cells, resulting in increased *Ren* and *Cd38* mRNAs, which leads to hypertension [85]. Consistent with this result, RAS activity in SAS patients has been found to be increased, which may elevate blood pressure [78,86]. In contrast, some studies have reported that Ren activity is not different between SAS patients and controls [87,88], suggesting that the blood pressure in SAS patients may not be determined by RAS.

Catecholamines are another determinant in blood pressure control. Therefore, we investigated the expression(s) of catecholamine-synthesizing enzymes using human and mouse neuroblastoma cells. We prepared cellular RNA from catecholamine-producing mouse Neuro-2a and human NB-1 neuroblastoma cells exposed to IH for 24 h, which produce catecholamines, and measured the RNA levels of *tyrosine hydroxylase* (*Th*), *L-3,4-dihydroxyphenylalanine* (*DOPA*) *decarboxylase* (*Ddc*), *dopamineβ-hydroxylase* (*Dbh*), and *phenylethanolamine N-methyltransferase* (*Pnmt*) by real-time RT-PCR. Real-time RT-PCR revealed that IH exposure significantly increased the mRNAs of *DBH* and *PNMT* in NB-1 and Neuro-2a cells. Immunoblot showed that the expression of DBH and PNMT in NB-1 cells was significantly increased by IH. Promoter assays using firefly luciferase as a reporter revealed that the promoter activities of *DBH* and *PNMT* were not increased by IH, suggesting that the gene expression of *DBH* and *PNMT* in response to IH was not regulated by transcription. Considering a miRNA-mediated posttranscriptional mechanism, the level of miR-375, which was found to target both *DBH* and *PNMT* mRNAs by the MicroRNA.org program, in IH-treated cells was measured by real-time RT-PCR, and the level of miR-375 was found to be decreased significantly compared with that of normoxia-treated cells. The IH-induced upregulation of *DBH* and *PNMT* was abolished by the introduction of the miR-375 mimic, but not abolished by the miR-375 mimic non-specific control RNA. These results indicate that IH increases DBH and PNMT levels via the inhibition of miR-375-mediated mRNA (*DBH* and *PNMT* mRNAs) degradation [89]. Therefore, it is inferred that in SAS patients, the upregulation of *DBH* and *PNMT* in neural cells in the adrenal medulla may induce hypertension, while miR-375 could play a crucial role in regulating the gene expression of *DBH* and *PNMT*. Taken together, IH upregulates the mRNA levels of *Ren* and *Cd38* in JG cells via the downregulation of miR-203-mediated mRNA degradation and the mRNA levels of *DBH* and *PNMT* in neuroblastoma cells via the inhibition of miR-375-mediated mRNA degradation, which may contribute to hypertension (Figure 3).

There are numerous reports of increased catecholamine synthesis, catecholamine secretion, and the upregulation of *PNMT* gene by IH in SAS patients and experimental models [90,91]. In catecholamine secretion from neuronal/adrenal chromaffin cells in hypoxia, hypoxia stimulates catecholamine secretion and/or gene expression in vitro and in vivo [92,93,94]. Previous studies of enzymic activities for catecholamine biosynthesis in the adrenal medulla, which is derived from the spontaneously hypertensive rat (SHR) neural crest, showed inconsistent results, as catecholamine biosynthetic enzyme activity was reported to be decreased [95,96,97], unchanged [98], and increased [99,100] in young SHRs. Ddc, Dbh, and Pnmt activities were decreased [96,101] or increased [102,103] in young SHRs, but unchanged in adult SHRs [96,103]. Due to the apparent correlation of catecholamine biosynthetic enzyme activities, some authors have suggested the possibility that the genes (*Ddc*, *Dbh*, and *Pnmt*) are co-regulated by a single locus [104]. Thus, the effect of IH on the expression of catecholamine synthesis is controversial, and it is important and necessary to clarify the relationship between IH and hypertension using experimental cellular or animal models and clinical samples in the future.

Cardiovascular dysfunction is one of the most common complications in patients with SAS [105]. Cardiovascular dysfunctions can occur as a complication of diabetes and/or hypertension [106]. Diabetes and hypertension can be caused by IH in SAS patients, as described above [5,7,17,64,85,89]. The CD38-cADPR signal system has been reported to be important for mammalian cell functions [107], including cardiac functions [108,109,110,111]. Most recently, we investigated the effects of IH on the gene expressions of the components of the Cd38-cADPR signal system (Cd38, ryanodine receptor 2 [Ryr2], that is predominantly expressed in cardiomyocytes, and FK506-binding protein 12.6 [Fkbp12.6], that is co-expressed with Ryr2, which works as a stabilizer of the Ryr2 Ca^2+^ channel, and dissociates from Ryr2 when cADPR is binded to [112]) in rat and mouse cardiomyocytes using an in vitro IH system. Real-time RT-PCR and western blot showed that the expressions of Cd38, Ryr2, and Fkbp12.6 in IH-treated cells were significantly reduced. In contrast, the expression of phosphatase and tensin homolog deleted from chromosome 10 (Pten) was upregulated by IH stimulation. The introduction of siRNA into Pten abolished the IH-induced reduction of Cd38, Ryr2, and Fkbp12.6 [113]. We treated H9c2 rat cardiomyocytes with 3-deaza-cADPR, a cell-permeable cADPR agonist [114], and introduced it into the cardiomyocyte culture medium, and found that the IH-induced decreases in Cd38, Ryr2, and Fkbp12.6 were attenuated, indicating that IH indeed reduces Cd38 expression, Cd38 activity (ADP-ribosyl cyclase activity for cADPR synthesis), and cADPR concentration in cardiomyocytes. The reduced cADPR concentration, in turn, decreased the recipients of cADPR (downstream workers in the signal system), Ryr2, and Fkbp12.6 in cardiomyocytes. These results indicate that cardiomyocyte dysfunction could be caused directly by IH itself, but not by diabetes and/or hypertension induced by IH. Recently, Dr. Nagata and his colleagues found that the treatment of (PB-7-23-111′1′4)-oxodiperoxy(2-pyridinecarboxylato-κN^1^,κO^2^)-vanadate(2-), dipotassium, dihydrate (a Pten inhibitor), and bisperoxovanadium-pic ameliorated left ventricular inflammation, fibrosis, and diastolic dysfunction in DS/Obese (DahlS.Z-lepr^fa^/Lepr^fa^) rats [115]. These reports strongly suggest that inhibitors of Pten maintain cardiac cell function and may serve as new drugs for cardiomyocyte function.

## 5. The Relationship between Metabolic Dysfunction and Dementia

Dementia is one of the most important health issues of global concern, comprising Alzheimer’s disease (AD), vascular dementia, Lewy body dementia, and other neurodegenerative and metabolic diseases. It is estimated that the number of people with dementia will increase from 57.4 million cases globally in 2019 to 152.8 million cases in 2050 [116]. AD is the most prevalent type of dementia and accounts for up to 70% of all cases of dementia, followed by vascular dementia, which accounts for 15–20% of all cases [117]. AD is characterized primarily by impaired episodic memory, which is often accompanied by numerous cognitive impairments in areas such as executive function, language, visuospatial abilities, and decision making [118]. Much observational and experimental evidence to date indicates that AD is caused by the accumulation of extracellular neuritic plaques composed of amyloid-β (Aβ), leading to the progressive accumulation and aggregation of hyperphosphorylated tau and the formation of intraneuronal neurofibrillary tangles (NFT) [119]. These two neuropathological hallmarks, Aβ and NFT, culminate in progressive and irreversible cognitive decline in AD patients [120,121]. Because the pathogenesis of obesity is complex and multifactorial, many studies have investigated the various biochemical and cellular mechanisms involved in obesity-induced cognitive dysfunction. A 36-year longitudinal study showed that middle-aged and older adults with both obesity and high abdominal circumference have a 3.6-fold increased risk of dementia, even after controlling for diabetes and other vascular comorbidities [122]. The underlying mechanism of obesity-induced dementia is reported to involve peripheral inflammation and increased blood–brain barrier (BBB) permeability to a neuroinflammatory response due to alterations in the gut microbiota composition and excessive white adipose tissue (WAT), where adipocytes that are increased in size secrete pro-inflammatory adipokines such as leptin, TNF-α, IL-1β, and IL-6, and affect cerebrovascular function [123]. Regarding the gut microbiota, a high-fat diet (HFD) in rats is linked to increased Firmicutes phyla, decreased Bacteroidetes phyla abundance, and increased serum lipopolysaccharide (LPS) [124]. The abundance of Bacteroidetes was positively associated with object recognition memory in diet-induced obesity mice [125]. HFD also increased Oscillibacter, which is a gram-negative anaerobic species that contains LPS, in mice fecal microbiota [126]. It is speculated that LPS can alter the structure of tight junction proteins, thus increasing gut permeability [127]. Through such a possible mechanism, LPS enters the bloodstream, leading to peripheral and central inflammation. Kreutzer et al. also demonstrated increased inflammation in the mediobasal hypothalamus in human adults with obesity, which was inversely correlated with *Parasutterella* sp. (Proteobacteria) and Marinilabiliaceae (Bacteroidetes) [128]. Thus, alterations in the composition of gut microbiota might influence cognitive function through inflammation. Consistent with such an inflammation hypothesis, many studies have also shown that inflammatory markers are present in the brain and blood of AD patients postmortem. The activation of astrocytes and microglial cells releases inflammatory cytokines, such as IL-1β, TNF-α, and IL-6, which trigger tau hyperphosphorylation [129]. Pistell et al. demonstrated that mice fed a very high-fat lard diet showed cognitive impairment and a significantly high expression of inflammatory markers, such as IL-6 and TNF-α, in their brains, suggesting a clear association between cognitive impairment and increased brain inflammation [130].

An increasing number of studies indicate that patients with type 2 DM are also between 1.5 and 3 times more likely to develop AD or vascular dementia [131]. Although the pathophysiological mechanisms of the increased risk of dementia due to the presence of DM are diverse and controversial, central insulin resistance, inflammation, and oxidative stress have mainly been proposed as etiologic mechanisms [132]. Insulin receptors, insulin-like growth factor (IGF)-1 receptors, and their post-receptor signaling partners are distributed throughout the brain [133]. Impaired insulin signaling has been reported to affect the brain in addition to peripheral tissues, as indicated by a decrease in protein kinase B phosphorylation in various animal and cell models of insulin resistance [134], and hyperglycemia can cause alterations in intracerebral insulin signaling pathways by protein glycosylation, which could result in an increase in Aβ deposits and phosphorylation of tau proteins [135]. A previous animal study demonstrated that a decrease in the antioxidant activity of the mitochondria due to aging or alteration in insulin/IGF-1 signaling by diabetes could increase the vulnerability to AD, since insulin prevents the decrease in oxidative phosphorylation and reduces oxidative stress induced by Aβ [136]. Insulin also stimulates mitochondrial protein synthesis, and IGF-1 prevents hyperglycemia-induced oxidative stress, and the insulin/IGF-1 signaling defects make neurons more vulnerable to reactive oxygen species [136]. Ruegsegger et al. indicated that insulin is a pivotal regulator of brain mitochondrial function and ATP production, and that insulin resistance adversely increases oxidative stress [137]. Because of the extensive overlap in signaling between insulin, neurotransmission, and neuroinflammatory responses, an ongoing area of research is identifying the genetic and/or epigenetic factors shared between AD and type 2 DM [138]. Furthermore, the increased risk of microvascular infarction associated with hyperglycemia contributes to the increased risk of vascular dementia [139,140]. Deane et al. also showed that, through interaction with receptors for advanced glycation end products holding cells in the vessel wall, Aβ crosses the BBB and expresses inflammatory cytokines and endothelin-1, the latter of which mediates vasoconstriction by Aβ in genetically manipulated mice [141]. Notably, hypoglycemia is also recognized as a risk factor for dementia, as it has been reported that severe hypoglycemia episodes are associated with significant neuronal damage in both the cortex and hippocampus in streptozotocin-induced diabetic rats [142]. Several studies have demonstrated that hypoglycemic people have twice the risk of dementia progression as those without hypoglycemic episodes [143]. Therefore, preventing hypoglycemia is just as important as preventing hyperglycemia in the treatment of diabetes to prevent dementia in diabetic patients. Regarding the mechanism of hypoglycemia-induced dementia, He et al. showed that recurrent moderate hypoglycemia disturbs mitochondrial morphology and function in the hippocampus by inhibiting the transient receptor potential canonical channel, TRPC6, leading to neuronal death and cognitive impairment in diabetic mice [144].

DM adversely affects oligodendrocytes, astrocytes, and microglia in the brain [132]. Conditions of diet-induced obesity or dysregulated metabolism lead to multiple pathologic changes in glia, including oligodendrocyte loss and impaired myelination [145], alterations in astrocyte autophagy [146], release of neurotransmitters [147], and activation of microglia [148]. These pathological changes may impair glia–neuron metabolic interactions and disrupt the energy supply chain to neurons, resulting in neuronal damage that may lead to cognitive impairment [132].

Hypertension is recognized as an important risk factor for dementia in old age, including AD and vascular dementia [149]. Hypertension is associated with brain atrophy and an increased number of Aβ plaques and NFTs in neocortex and hippocampus in an autopsy study [150]. The proposed mechanisms or mediators of risk linking hypertension with cognitive impairment include cerebral ischemia and cerebral hemorrhage, which result in structural changes in the brain, chronic kidney disease, extracranial large vessel atherosclerosis, and cardiac disease. Moreover, it is also proposed that an inflammatory mechanism disrupts the BBB with microglial activation and impairs the glymphatic clearance of Aβ [151].

## 6. The Relationship between IH and Aβ Production/Degradation

In recent years, the association of OSA with neurodegenerative disorders and the early onset of cognitive decline has been highlighted [152,153,154,155]. Cognitive impairment is reported to be a comorbidity of OSA [156], and patients with a clinical diagnosis of AD are even more likely to develop SAS [157]. A recent meta-analysis found that AD patients have a 5-fold higher risk of presenting OSA than age-matched controls, and that approximately 50% of AD patients experience OSA after their initial diagnosis [157]. Conversely, OSA may promote the aggravation of existing AD, as untreated SAS is associated with cognitive decline and conversion to AD at younger ages [158]. Furthermore, a recent clinical study demonstrated that individuals who are older than 65 years, and have untreated SAS, presented greater amyloid deposition mainly over the posterior cingulate, cuneus, and precuneus areas, which are typically altered in AD, even in the absence of cognitive or behavioral manifestations [159].

AD is a neurodegenerative disease characterized by the presence of Aβ plaques, which are extracellular aggregates of misfolded amyloid precursor protein (APP) and NFTs. The accumulation of Aβ plaques and NFTs corresponds to neuronal degeneration and death observed in various areas of the brain [160]. The induction of the amyloidogenic pathway involves the cleavage of APP by β-secretase, mainly beta-site amyloid precursor protein cleaving enzyme-1 (BACE1), first generating soluble APPβ peptide. Subsequently, γ-secretase cleaves the C-terminal fragment of APP to generate neurotoxic Aβ peptides. Although Aβ of various lengths are formed, the major component of plaque formation is Aβ_1–42_, which has a highly hydrophobic C-terminus that initiates the aggregation of Aβ, oligomerizing to form higher-order insoluble structures that diffuse throughout the brain [161,162]. On the other hand, Aβ generation can be prevented via a non-amyloidogenic APP processing pathway that is mediated by α-secretase and γ-secretase [161].

There are several animal and clinical studies that have investigated the effect of IH on Aβ deposits. Shiota et al. showed that chronic IH (alternating 5% O_2_ and 21% O_2_ every 10 min for 8 h per day during daytime for 8 weeks) significantly increased levels of Aβ_42_ in the brains of AD transgenic mice, as well as intracellular Aβ in the brain cortex [163]. Bu et al. demonstrated that serum Aβ and phosphorylated tau 181 levels were significantly increased in OSAS patients [164]. Recently, Bhuniya et al. also reported that serum levels of Aβ and total tau were higher in OSAS patients than in controls, but serum levels of phosphorylated tau were not changed significantly [165]. In contrast, Yue et al. demonstrated that IH (14.3% O_2_ 4 h/day for 14 d or 28 d) alleviates memory impairment and reduces Aβ accumulation and inflammation in the brain of double transgenic mice expressing a chimeric mouse/human APP (Mo/HuAPP695swe) and a mutant human presenilin 1 (PS1-dE9) with the reduction of IL-1β, IL-6, and β-secretase levels [166]. Macheda et al. also indicated no effect of IH (10 h/day for 28 d) on Aβ levels or plaque load in AD-relevant mice (APP/PS1 KI) [167]. Regarding the involvement of astrocytes in response to IH, in addition, they showed a significant increase in glial fibrillary acidic protein (GFAP) staining in the APP/PS1 KI mice following IH exposure, but not in the wild type mice, suggesting that the presence of Aβ pathology can have an additive effect through IH exposure on reactive gliosis [167]. For these conflicting results, consideration should be given to the severity of hypoxia within episodes, the duration of hypoxic episodes, the number of hypoxic episodes per day, the pattern of presentation across time (e.g., within vs. consecutive vs. alternating days), and the cumulative time of exposure that would lead to differences in the extent to which oxidative stress is produced. Severe/chronic IH protocols tend to be pathogenic, whereas any beneficial effects are more likely to arise from modest/acute IH exposures, as severe hypoxia (2–8% inspired O_2_) and more episodes per day (48–2400 episodes/day) elicit progressively greater pathology [168].

BACE1 is considered a potential and key regulator in the progression of AD because of its involvement in the rate-limiting step that produces the toxic Aβ_1–42_ peptide. Although BACE1 mRNA and protein are found at low levels, even in many cell types, they are mainly expressed in the brain and pancreas [169]. Peroxisome proliferator-activated receptor gamma coactivator 1-alpha (PGC-1α) reduces BACE1 levels by repressing transcription in the *BACE1* promoter region [170]. Although the link between IH and BACE1 has not been fully elucidated, Li et al. investigated the mRNA levels of *BACE1* and *PGC-1α* by RT-PCR in the rat carotid body after chronic IH exposure (alternating 10% O_2_ and 21% O_2_ every 6 min for 8 h/day from 8:30 to 16:30 for 2 weeks), showing that chronic IH reduced the mRNA level of *BACE1*, but augmented the mRNA level of *PGC-1α* [171]. On the other hand, Ng et al. found that even short-term chronic IH exposure (3 d) caused significant increases in the generation of Aβ peptides and in the expression of BACE, presenilin, and HIF-1α protein levels in the hippocampus of rats. Moreover, the chronic IH-induced hippocampal Aβ peptide generation could be abolished by a daily pharmacological administration of melatonin (10 mg/kg), which reduced the BACE, but not presenilin expression [172]. Thus, there is no settled opinion on the effect of IH on the expression of BACE-1, depending on the condition of IH exposure.

γ-Secretase is a transmembrane protein complex containing presenilin (PS), nicastrin, anterior pharynx defective-1, and presenilin enhancer-2, which are sufficient for γ-secretase activity [173]. PS1 is the catalytic subunit of γ-secretase [174]. Although the effect of IH on γ-secretase has rarely been examined, Ryou et al. showed that IH (5–8 daily cycles of 5–10 min of 9.5–10% fractional inspired O_2_ + 4 min of 21% fractional inspired O_2_) attenuates ethanol-withdrawn induction of PS1 protein, but not *PS1* mRNA, and prevents cerebrocortical Aβ_40_ and Aβ_42_ protein accumulation in ethanol-withdrawn rats [175].

Several Aβ-degrading proteases have been identified, such as neprilysin (NEP), insulin-degrading enzyme, endothelin-converting enzyme, angiotensin-converting enzyme, plasminogen activators, or different matrix metalloproteinases [176]. However, little has been elucidated about the effect of IH on the expression of these enzymes. Oliveira et al. showed that a six-week IH increased NEP expression and activity selectively in the temporal cortex, but not in the hippocampus and frontal cortex, and that the increase in NEP activity and expression was reverted, followed by two weeks recovery in normoxia [177]. Further cellular and animal studies will be needed to understand the impact of IH on Aβ production and degradation.

## 7. Conclusions

Recently, there has been great medical and scientific interest in the association of IH, a hallmark manifestation of SAS, with metabolic dysfunction and dementia. Cellular studies have indicated that IH influences the expression of various genes related to metabolism in various cells, and that IH may induce decreased appetite, impaired glucose tolerance, insulin resistance, hypertension, and cardiovascular dysfunction.

IH is strongly associated with dementia via different mechanisms, including insulin resistance, inflammation, and ischemia. However, the direct effect of IH on the development or exacerbation of cognitive impairment remains unclear (Figure 4). To further understand the complex relationship between IH and dementia, more molecular, clinical, and translational research in vitro and in vivo is required.

## Figures and Tables

**Figure 1 ijms-23-12957-f001:**
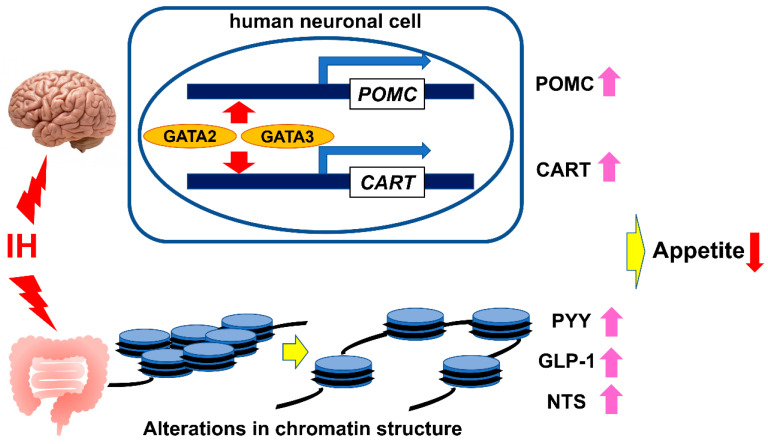
Schematic diagram of the effect of IH on the gut–brain axis with respect to appetite regulation. IH may have anorexigenic effects on the gut–brain axis by upregulating *POMC* and *CART* mRNAs via GATA transcription factors (GATA-2 and GATA-3) in neuronal cells [35], and by upregulating *PYY*, *GLP-1*, and *NTS* mRNAs via an alteration in chromatin structures of the *PYY*, *GLP-1*, and *NTS* genes in enteroendocrine cells [42].

**Figure 2 ijms-23-12957-f002:**
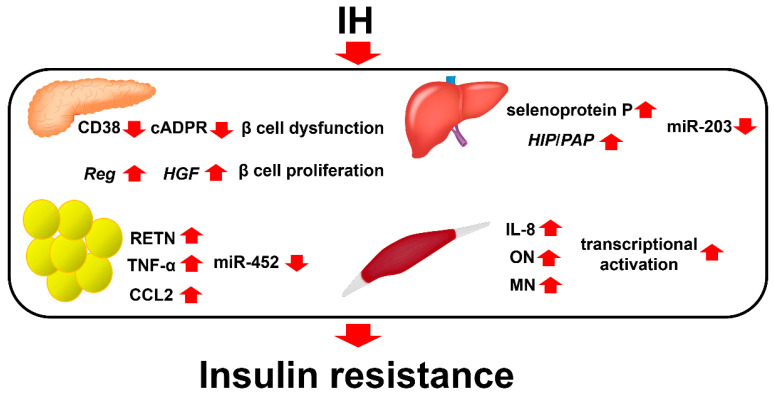
The relationship between IH and diabetes (insulin resistance). IH is involved in the reduction of glucose-induced insulin secretion from pancreatic β cells via downregulation of CD38 [5] and pancreatic β cell replication by the upregulation of Reg I and hepatocyte growth factor in pancreatic β cells [62]. IH also involves the upregulation of selenoprotein P and HIP/PAP in hepatocytes via downregulation of miR-203 [17]; upregulation of adipokines such as CCL2, TNF-α, and RETN in adipocytes via downregulation of miR-452 [64]; and upregulation of myokines such as IL-8, osteonectin, and myonectin in skeletal muscle cells [65,66,67], all of which can contribute to insulin resistance, glucose intolerance, and obesity [69].

**Figure 3 ijms-23-12957-f003:**
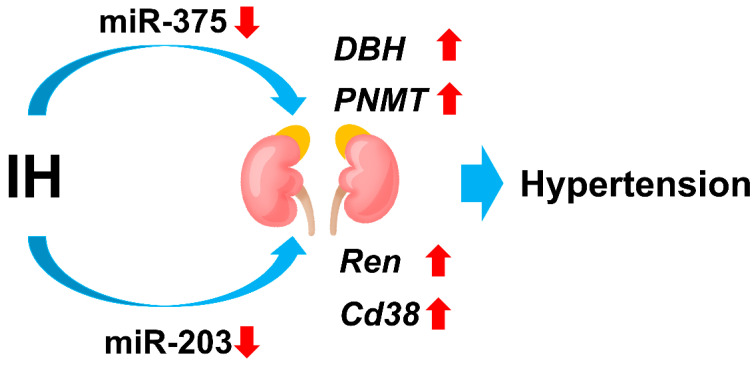
Schematic view of IH on the expression of *Ren* and *Cd38* in renal JG cells and *DBH* and *PNMT* in adrenomedullary neuroblastoma cells. IH upregulates *Ren* and *Cd38* mRNAs in renal JG cells by downregulating miR-203 [85] and *DBH* and *PNMT* mRNAs in adrenomedullary neuroblastoma cells by downregulating miR-375 [89].

**Figure 4 ijms-23-12957-f004:**
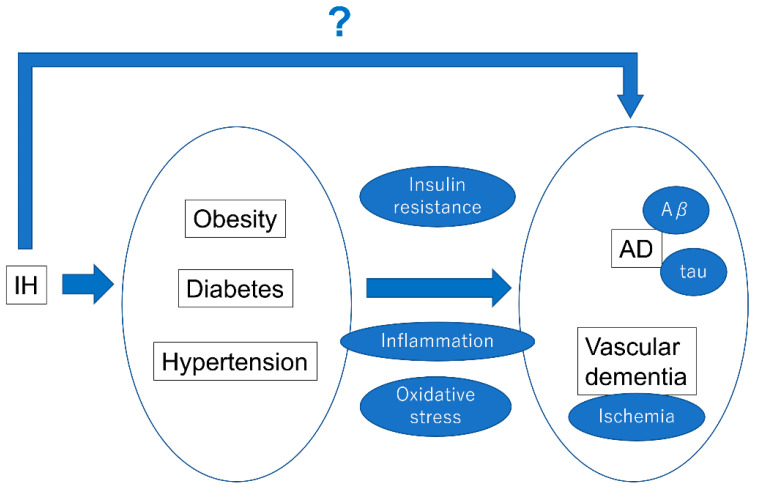
The relationship between IH, metabolic syndrome, and dementia. The contribution of IH to obesity, diabetes, and hypertension is becoming clearer and clearer. Moreover, evidence of the contribution of obesity, diabetes, and hypertension to dementia is growing. Future clarification of the direct effects of IH on the causative factors of dementia is awaited.

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
