# Peer review of "The Impact of Intermittent Hypoxia on Metabolism and Cognition"

_ijms, 2022, doi:10.3390/ijms232112957_

Round 1

Reviewer 1 Report

This is a very well written and interesting article. Well illustrated and, in principle, ready for publication in the desired journal. I have only a few suggestions to the authors.

1) I would like the authors to discuss the role of astrocytes in the regulation of processes occurring during hypoxia and sleep apnea syndrome

2) Conclusion makes sense to shorten

Author Response

Dear Dr. Wiranpat Olanwanit,

Re: ijms-1923014

Many thanks for your kind E-mail letter of October 7, 2022 along with the comments and suggestions, concerning the manuscript (ijms-1923014). We are grateful for your favorable review and helpful criticisms, which significantly encourage us to produce the revised version of the manuscript.

According to the reviewer's comments and your kind suggestion, the manuscript has now been suitably revised as follows:

Changes and responses to the Editor/reviewer's points:

Comments from the editor and reviewers:
-Reviewer 1

This is a very well written and interesting article. Well illustrated and, in principle, ready for publication in the desired journal. I have only a few suggestions to the authors.

#1) I would like the authors to discuss the role of astrocytes in the regulation of processes occurring during hypoxia and sleep apnea syndrome

<Response>

According to your kind suggestion, we added the literature as Ref. 168 in the REVISED VERSION to discuss the response of astrocytes by IH.

#2) Conclusion makes sense to shorten

<Response>

According to your constructive suggestion, we have shortened conclusion section to make it more readable and concise.

Comments from the editor and reviewers:

-Reviewer 2

This is a comprehensive review article describing the impact on IH or Metabolism and Cognition.  The article is well written, and the information is important since IH can occur in several disease processes, particularly in adults with sleep-disordered breathing and IH contributes to the co-morbidities that occur in adults with sleep-disordered breathing.

My primary critique (which I believe is major) is that many of the findings discussed are from their own work, and even then, it was difficult to tell whether the work they were referring to is published or unpublished.

<Response>

Thank you for your meaningful remarks. All the findings we are discussing are based on already published articles that can be searched on Pubmed.

Broad statements were made based on their findings in previous studies that I believe were in vitro/ preclinical studies, and there was no indication of sample sizes or translational capacities, so it's difficult to know whether it’s a stretch or not unless I read their previous work.

<Response>

Although our findings are indeed in vitro experiments, we have elucidated the molecular mechanisms of IH-induced changes in relevant gene expression, suggesting clinical interpretation that IH can cause metabolic disorders. In addition, “Aim” of IJMS is “The International Journal of Molecular Sciences (ISSN 1422-0067; CODEN: IJMCFK; ISSN 1661-6596 for printed edition) provides an advanced forum for molecular studies in biology and chemistry, with a strong emphasis on molecular biology and molecular medicine. Our aim is to provide rigorous peer review and enable rapid publication of cutting-edge research to educate and inspire the scientific community worldwide.” Therefore, we think molecular mechanisms of IH-induced changes in relevant gene expression considered the molecular mechanism of SAS to be suitable for the purposes of IJMS, and gave priority to it.

I also found the first several sections with all the references to signaling pathways in all the different cells. cells quite dense 

<Response>

We have extensively studied IH-induced changes in gene expression associated with metabolic syndrome using various cells, and think that all the molecular mechanisms described here are important to understand molecular aspects of IH and metabolic disorders.

The best part of the review is sections 5 and 6. I recommend that those sections come first, and the earlier sections should be edited to improve readability and be more focused on the brain.  The first part should focus on the effects of metabolic disorder and IH on signaling pathways in neuronal cells and then link it to the later sections.  The information about all the other cells is distracting and makes the review somewhat disjointed.  A stronger focused review of metabolic disorders with and without IH and the development of cognitive impairment would be an excellent addition to the published literature.

<Response>

We are glad that you are interested in the mechanism of onset of dementia due to metabolic disorders and IH, and thank you for your kind suggestion. While there is much evidence that metabolic syndrome is strongly correlated with dementia, we have extensively studied IH-induced changes in gene expression associated with metabolic syndrome using various cells, and we are interested in how IH may directly contribute to the development of dementia. The purpose of this review is to summarize in vitro findings on the effects of IH on metabolic abnormalities, mainly based on our original findings, and to spotlight the unexplored field of IH-induced dementia onset. Therefore, we think it is better to keep the section order by first outlining our original molecular findings on the pathogenesis of metabolic syndrome due to IH, and then, as a development, discussing the relatively unexplored impact of metabolic syndrome and IH on the development of dementia.

All the changes described above are in RED in the REVISED VERSION.

We should like to thank the reviewers for the helpful comments and hope that we have now produced a more balanced and better account of our work. We hope that the revised version of the manuscript is acceptable to the Editor and will be followed by publication in International Journal of Molecular Sciences as soon as possible.

Thank you again for your kindness.

Sincerely,

Ryogo Shobatake, MD & PhD

Department of Neurology,

Nara Medical University School of Medicine

840 Shijo-cho, Kashihara, Nara, 634-8522, Japan

TEL: +81-744-22-3051 ext. 3417

FAX: +81-744-24-6065

Reviewer 2 Report

This is a comprehensive review article describing the impact on IH or Metabolism and Cognition.  The article is well written, and the information is important since IH can occur in several disease processes, particularly in adults with sleep-disordered breathing and IH contributes to the co-morbidities that occur in adults with sleep-disordered breathing. 

My primary critique ( which I believe is major)  is that many of the findings discussed are from their own work, and even then, it was difficult to tell whether the work they were referring to is published or unpublished. Broad statements were made based on their findings in previous studies that I believe were in vitro/ preclinical studies, and there was no indication of sample sizes or translational capacities, so it's difficult to know whether it’s a stretch or not unless I read their previous work.

I also found the first several sections with all the references to signaling pathways in all the different cells. cells quite dense   The best part of the review is sections 5 and 6. I recommend that those sections come first, and the earlier sections should be edited to improve readability and be more focused on the brain.  The first part should focus on the effects of metabolic disorder and IH on signaling pathways in neuronal cells and then link it to the later sections.  The information about all the other cells is distracting and makes the review somewhat disjointed.  A stronger focused review of metabolic disorders with and without IH and the development of cognitive impairment would be an excellent addition to the published literature.   

Author Response

(The authors gave the same response as above.)
